# Seroprevalence of Varicella in Pregnant Women and Newborns in a Region of Colombia

**DOI:** 10.3390/vaccines10010052

**Published:** 2021-12-31

**Authors:** Viviana Lenis-Ballesteros, Jesús Ochoa, Doracelly Hincapié-Palacio, Alba León-Álvarez, Felipe Vargas-Restrepo, Marta C. Ospina, Seti Buitrago-Giraldo, Francisco J. Díaz, Denise Gonzalez-Ortíz

**Affiliations:** 1Hector Abad Gómez National Faculty of Public Health, Universidad de Antioquia UdeA, Medellin 050010, Colombia; jesus.ochoa@udea.edu.co (J.O.); doracelly.hincapie@udea.edu.co (D.H.-P.); albaluz3105@gmail.com (A.L.-Á.); felipe.vargasr@udea.edu.co (F.V.-R.); 2Laboratory of Public Health of the Regional Secretariat of Health and Social Protection of Antioquia, Medellin 050010, Colombia; martha.ospina@antioquia.gov.co (M.C.O.); seti.buitrago@gmail.com (S.B.-G.); 3School of Medicine, Universidad de Antioquia UdeA, Medellin 050010, Colombia; francisco.diaz@udea.edu.co; 4Secretariat of Health of Medellin, Medellin 050010, Colombia; ld.gonzalez.ortiz@gmail.com

**Keywords:** varicella, pregnant women, seroepidemiological studies, vaccine preventable disease, Colombia

## Abstract

We estimate the seroprevalence of IgG antibodies to varicella zoster virus (VZV) based on the first serological study in a cohort of pregnant women and newborns from the Aburrá Valley (Antioquia-Colombia) who attended delivery in eight randomly chosen hospitals. An indirect enzyme immunoassay was used to determine anti-VZV IgG antibodies. Generalized linear models were constructed to identify variables that modify seropositivity. In pregnant women, seropositivity was 85.8% (95% CI: 83.4–85.9), seronegativity was 12.6% (95% CI: 10.8–14.6), and concordance with umbilical cord titers was 90.0% (95% CI: 89–91). The seropositivity of pregnant women was lower in those who lived in rural areas (IRR: 0.4, 95% CI: 0.2–0.7), belonged to the high socioeconomic status (IRR: 0.4, 95% CI: 0.2–0.7), and had studied 11 years or more (IRR: 0.6, 95% CI: 0.4–0.8). Among newborns, seropositivity was lower in those who weighed less than 3000 g (IRR: 0.8, 95% CI: 0.6–1.0). The high seropositivity and seronegativity pattern indicates the urgent need to design preconception consultation and vaccination reinforcement for women of childbearing age according to their sociodemographic conditions, to prevent infection and complications in the mother and newborn.

## 1. Introduction

Varicella is a global infectious disease, caused by primary infection with the varicella zoster virus (VZV), that remains latent in the nervous system; it can reactivate years after infection and generate complex neurological diseases (herpes zoster) in adults, depending on the behavior of the host’s immune system [1,2]. It is not a trivial disease; it has complications in pregnant women and in newborns, high transmissibility in the exposed susceptible individuals, and has secondary attack rates above 70% [3].

It can affect pregnant women; 4 to 7 of every 10,000 pregnant women are estimated to have a primary infection with VZV causing congenital varicella syndrome (CVS) if vertical transmission occurs in the first two trimesters of pregnancy. SVC and multisystem involvement are associated with 30% mortality and intrauterine growth retardation [4]. Neonatal varicella, which has a high probability of infecting the child, occurs if the infection of the pregnant woman takes place a few days before or during the two months after delivery [5].

Since the 1960s, with greater momentum at the beginning of the 21st century [6], seroprevalence studies have analyzed disparities in exposure to VZV according to the climate (temperate or tropical), the socioeconomic conditions of developed and underdeveloped countries [7,8], and the feasibility of introducing childhood vaccination and other preventive measures. Seronegativity has been reported from 4% to values above 30%, with higher figures in recent years, possibly due to changes in exposure to the virus, indicating the need to focus prevention strategies on women of childbearing age [9,10,11,12].

The seroepidemiological behavior of varicella has been explored in different regions. In the 2000s, it was reported that the levels of antibodies to VZV in Latin American populations, coming from Brazil, Argentina, Chile, the Dominican Republic, and Mexico, presented significant differences according to sex, with the seroprevalence remaining higher in women than in men. However, the proportion of IgG antibodies to varicella did not exceed 88%. In Venezuela, a seroprevalence for this disease of 75.9% was reported without differences in the population studied according to sex [13].

In Colombia, 68.8% prevalence of VZV antibodies was reported in 2006, calculated for a sample stratified by socioeconomic levels of 1163 people between 1 and 39 years of age; seropositivity for antibodies against VZV was 50% in children aged 1 to 4 years and 86.7% in the group aged 31 to 39 years [14].

The exploration of VZV seroprevalence in American countries points to some common conclusions: Women and younger age cohorts have a higher risk of contracting the disease because they have lower rates of VZV seroprevalence. These studies recommend prioritizing immunization of susceptible people at high risk of complications such as health professionals, family contacts of immunosuppressed patients, and non-pregnant women of childbearing age [13,15].

In Latin America and the Caribbean, seroprevalence studies are necessary, as they are considered the most reliable assessment of the degree of viral exposure [16]. In this region, with its biogeographic and socioeconomic diversity, there is a high risk of VZV transmission, especially in tropical areas. In addition, there are limitations in the registration of cases of varicella and herpes zoster, which leads to a lack of knowledge of infection and disease patterns [8]. Environmental factors, socioeconomic conditions, migration, and cultural practices seem to influence the differences in rates of exposure to VZV and consequent seroprevalence [17]. The age of the pregnant women, area of residence, number of deliveries, and occupation have been reported as conditions that influence the seroprevalence of varicella in different regions of the world [18,19].

There are no known published studies on seroprevalence in pregnant women or the transfer of antibodies to the newborn to guide the development of preventive programs for maternal, congenital, and neonatal varicella in Colombia and Latin America [8].

Universal vaccination for VZV has been gradually included in national plans [8]. In Colombia, since June 2015, this vaccine has been included in the Expanded Program on Immunization (EPI), with a single dose for children aged one year old, covering the cohort born after 1 July 2014. In June 2019, the application of a booster at five years of age was established as a national guideline [20].

With these considerations, this study focused on estimating the seroprevalence of IgG for varicella in pregnant women and newborns from the Aburrá Valley in Antioquia-Colombia, in a pre-vaccination era, as a contribution to the knowledge of seropositivity distribution by socioeconomic and demographic variables, and perhaps to the coordination and design of prevention measures for maternal and neonatal VZV infection.

## 2. Materials and Methods

### 2.1. Design and Data Sources

This cross-sectional study used data derived from an investigation within the departmental serosurveillance program [21]. Samples were collected from a random selection of pregnant women who attended delivery in eight hospitals in Aburrá Valley, Department of Antioquia, Colombia. Peripheral venous blood samples of the pregnant woman and umbilical cord samples were collected to estimate the seroprevalence of IgG antibodies to VZV, the concordance of the levels of antibodies in the mother and in the umbilical cord, and the distribution of seropositivity according to sociodemographic variables and gestation recorded before delivery.

### 2.2. Population and Sample

The study population comprised pregnant women residing in the Aburrá Valley (Antioquia, Colombia) who attended hospital delivery care. The selection of included hospitals was conducted using probabilistic, multistage sampling, stratified by conglomerates.

### 2.3. Inclusion and Exclusion Criteria

Pregnant women at 37 or more weeks of gestation, who agreed to participate in an informed and voluntary manner, were included. Pregnant women with multiple gestation, an infectious process 72 h before delivery, serious decompensated illnesses, or in advanced labor were excluded [21].

### 2.4. Lab Tests

The indirect immunoenzymatic Vircell^®^ test, with a specificity of 96% (95% CI: 79–99) and sensitivity of 96% (95% CI: 88–99), was used to determine IgG antibodies against VZV in human serum/plasma (Vircell Microbiologists, Granada-Spain). The antibody index was calculated by multiplying the ratio of the Optical Density (OD) of the sample to OD of the cut-off serum by 10 ((sampleOD/cutoffOD))x* (10)), resulting in three categories: Negative for antibody index values less than 9, equivocal for values of this index between 9 and 11, and positive for values greater than 11.

### 2.5. Statistical Analysis

Global seroprevalence levels were calculated, considering an expansion factor for the sample, which allowed inference to the population.

The characteristics of the pregnant women were described according to the categories of the IgG antibody level for VZV.

Count models (generalized linear models—negative binomial family) were used separately for the dependent variables “number of seropositive pregnant women” and “number of seropositive newborns”. This model allowed the verification of variables that modify the seropositivity count in pregnant women and newborns. The independent variables were the area of residence, socioeconomic status, the gestational week at the time of delivery, age, and years of study of the mother. The weight of the newborn was included as an independent variable in the newborn model. The percent change in seropositivity was calculated as follows: 1-IRR.

The antibody concentration of the pregnant woman and the cord were compared on a sunflower plot as an alternative to conventional scatter diagrams. Concordance coefficients and Lin’s 95% confidence intervals (95% CI) were quantified [22]. The kappa index and its 95% CIs (Jackknife simulation) [23] were also constructed using the categorized variables (positive-negative) of antibody titers.

Data processing was performed with Stata version 13 software (StataCorp, College Station, TX, USA).

## 3. Results

### 3.1. Description of the Study Population

A total of 799 maternal plasma samples and 751 umbilical cord samples were analyzed (Figure 1).

The pregnant women were 13 to 43 years old (median: 23 years; IQR: 20–28). The largest proportion of participants came from urban areas, had an educational level of 11 years or less, belonged to the low socioeconomic status, were affiliated with subsidized social security, and their households lived without overcrowding (Table 1).

Nearly half were primiparous, delivery was between 37 and 41 weeks of gestation, without presenting maternal morbidity (Table 1). Newborns had a similar distribution by sex, and birthweight was 3000 g or more in 64.31% (n = 483) (Table 1).

### 3.2. Seroprevalence

The global seroprevalence of IgG for varicella in pregnant women was distributed as seropositivity of 85.8% (95% CI: 83.4–85.9), seronegativity of 12.6% (95% CI: 10.8–14.6), and equivocal results of 1.6% (95% CI: 1.0–2.6).

Analysis was conducted on 751 samples of mother–umbilical cord pairs; 96.7% (95% CI: 95.1–97.8) of these pairs showed IgG seropositivity values for VZV. That is, for each seropositive mother, the sample from her umbilical cord was also seropositive.

The ratio of cord to pregnant female titers was 1.10 (95% CI: 1.09–1.12). The density distribution between the IgG antibody index of the pregnant woman and the umbilical cord is summarized in Figure 2. The 15 dark “flowers” show the region with the highest density (titers between 25 and 41) of antibody indices of the pregnant women and the umbilical cord (n = 355 observations). The 53 “light flowers” indicate a greater dispersion around the antibody values (n = 322 observations).

The concordance coefficient between the titers of the pregnant woman and the cord was 90.0% (Lin’s 95% CI: 89–91). The qualitative approach (titers > 11) quantified a kappa index (between the titers of the pregnant woman and the cord) of 90.7% (95% CI: 86.0–95.4).

The area of residence, the socioeconomic status, and the years of study of the pregnant women were found to have significant differences in the ratio of seropositivity rates, both for pregnant women and for the umbilical cord (Table 2). The seropositivity rate (IRR) was lower in mothers living in rural areas, of high socioeconomic status, and those who had studied 11 years or more (Table 2).

The percentage change in the seropositivity of pregnant women whose delivery was at 41 weeks of gestation was 50% lower (compared to maternal seropositivity in those whose delivery was attended at 37 weeks of gestation). The seropositivity in umbilical cord blood was 20% lower in those who weighed less than 3000 g (Table 2).

## 4. Discussion

Pregnant women from the Aburrá Valley of Antioquia presented a global seropositivity against varicella of 85.8% (95% CI: 83.4–85.9) and seronegativity of 12.6% (95% CI: 10.8–14.6).

These findings are consistent with seroprevalence studies in pregnant women that reported seropositivity figures higher than 80%. Plans et al., 2007, published seropositivity of 96.1% (95% CI 95.1–97.1) in a sample of 1522 pregnant women, representative of the population of pregnant women in Catalonia-Spain [9]. Karunajeewa and Col. reported, in 2001, seropositivity for varicella of 94% in pregnant women from Australia who attended a health service between 1998 and 1999 [24]. Ibrahim et al., between 2016 and 2017, detected seropositivity of 88.3% in pregnant women captured in the prenatal control of a university hospital in Egypt (n = 294) [18].

This study showed a positive and high concordance in the transfer of antibodies from the pregnant woman to the newborn. Van der Zwet et al. reported a high transfer of antibodies and a high correlation between the levels of IgG anti VZV of the newborn and the mother in a study carried out in 221 newborns and 43 mothers in 2002. These authors concluded that the maternal antibody titer was the main determinant of the newborn’s titer. In this study, gestational age had less influence on the level of the neonate’s titers [25]. These findings suggest that the vaccination of women of childbearing age could impact the protection of the newborn in the first months of life due to the IgG anti-VZV antibodies transferred by the mother [25,26].

The literature has reported a greater transfer of IgG antibodies for neonates who reached a gestational age greater than 28 weeks [25,26,27]. Although an efficient transfer of anti-VZV IgG antibodies is reported as gestational age increases, new research will be necessary to detail the impact of this transfer in term pregnancies with regional applicability in Colombia.

In this study, the ratio of seropositivity rates in pregnant women and the umbilical cord showed significant differences in some variables that reflect heterogeneity in exposure to VZV according to socioeconomic conditions. The high seropositivity found was similar to that reported in other tropical countries in the general population [28,29]. This perhaps indicates the need to resume the study of the theory of the explanation of the social determination of varicella transmission and the methods used to capture inequity in the distribution and occurrence of the disease. This work, like others, did not consider the social structure in the study population, which makes it unlikely to capture these inequities in exposure to VZV.

The age of the mother has been considered a VZV exposure variable in pregnant women studied in Egypt, Italy, and Spain, where older women have a greater probability of having IgG levels for VZV [9,12,18,30]. In the pregnant women from the Aburrá Valley, a similar distribution of seropositivity by age was observed in the pregnant women and umbilical cord samples. The vaccination strategies for varicella in Colombia within the EPI began in 2015 aimed at children under one year of age, not including women of childbearing age [20]. For this reason, the presence of maternal antibodies in this study may be due to natural exposure to the virus, perhaps due to the high incidence of varicella in Colombia.

The present study has several limitations. We do not know the history of disease or of vaccination for varicella in pregnant women. As has been established in the literature, antibody titers do not make any differentiation between natural or vaccine exposure prior to infection and a correlate of protection against the varicella-zoster virus has not been fully established [31,32]. We assume that the pregnant women in our study were not exposed to varicella vaccination in childhood, since universal vaccination aimed at children only began in 2015 in Colombia.

Due to ethical and cultural considerations, it was not possible to determine the presence of antibodies in the first months of life of newborns. The study was directed at healthy pregnant women of 37 weeks or more of gestation, which made it impossible to quantify the prevalence of antibodies in preterm newborns and in pregnant women with comorbidities.

This study provides the scientific basis for additional varicella prevention measure developments that reduce the risk of complications derived from primary infection during pregnancy in pregnant women and thus mitigate the risk of neonatal or congenital varicella syndrome [12]. These measures may include periodic studies of seroprevalence of pregnant women and newborns (hospital serosurveillance) as shown in this study and screening or vaccination reinforcement of women of childbearing age. Some studies have suggested the immunization of women of childbearing age as an immune booster for VZV [9,12], because universal childhood vaccination reduces (over time) the chances of natural exposure to the virus and increases the susceptible population in adults. This vaccination booster in women of childbearing age should be evaluated in Colombia, in terms of the cost benefit [18], the viability of the preconception consultation (including the investigation about varicella), and the age at the time of vaccine application, under the consideration that vaccination should be performed prior to pregnancy.

Education in routine fertility and prenatal follow-up programs, which involves considerations such as the risk of becoming ill in pregnancy, can positively impact the prevention of congenital and neonatal varicella [9,12,18].

## 5. Conclusions

High IgG seropositivity for VZV was detected in pregnant women as was an efficient transfer of antibodies to the newborn. However, the proportion of unprotected pregnant women is significant and indicates that it is necessary to design additional measures (preconception consultation and vaccination of women of childbearing age) to mitigate the risk of disease during pregnancy and the neonatal period. It is necessary to consider the influence of some conditions (Colombian and other lower-middle-income countries) such as the area of residence, the socioeconomic status, and schooling to guide the proposed measures.

## Figures and Tables

**Figure 1 vaccines-10-00052-f001:**
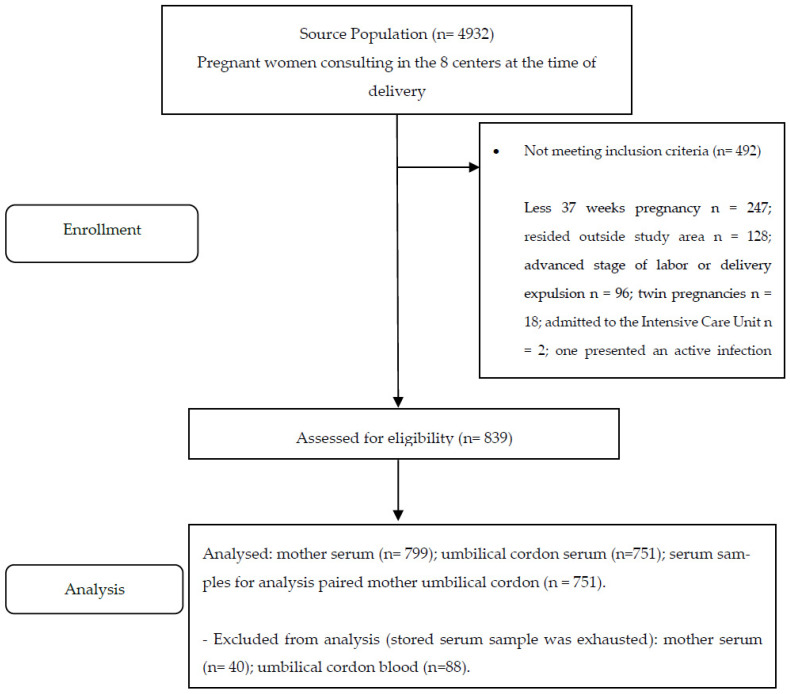
Flowchart of the study population. Seroprevalence of varicella in pregnant women and umbilical cord blood, Valle de Aburrá, Antioquia, Colombia.

**Figure 2 vaccines-10-00052-f002:**
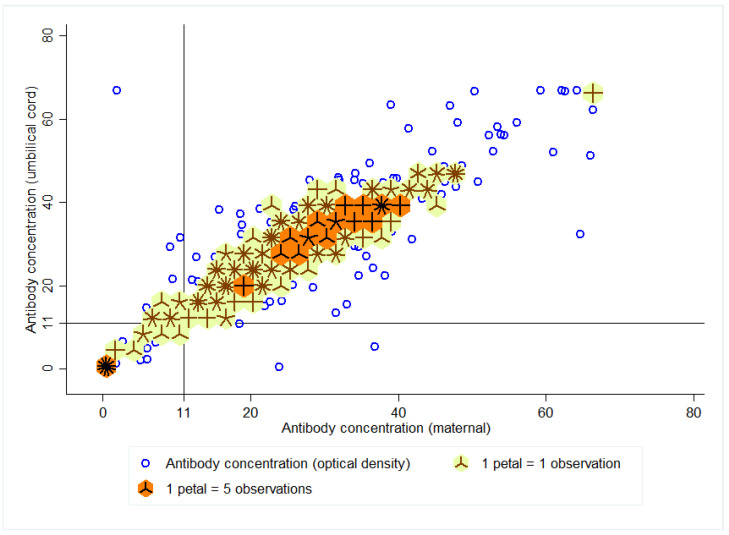
Density distribution between the anti-varicella IgG antibody index of the pregnant woman and the umbilical cord.

**Table 1 vaccines-10-00052-t001:** Seroprevalence status for varicella by characteristics of the pregnant woman and the newborn (umbilical cord). Aburrá Valley, Antioquia, Colombia.

**Pregnant Women**	**Total** **n (%)**	**Seropositive** **n (%)**	**Seronegative** **n (%)**	**Equivocal** **n (%)**
**Age (years)**				
13 to 22	362 (45.3)	298 (43.7)	60 (57.7)	4 (30.8)
23 to 32	341 (42.7)	299 (43.8)	36 (34.6)	6 (46.2)
33 to 43	96 (12.0)	85 (12.5)	8 (7.7)	3 (23.1)
**Area**				
Urban	760 (95.1)	651 (95.5)	98 (94.2)	11 (84.6)
Rural	39 (4.9)	31 (4.5)	6 (5.8)	2 (15.4)
**Educational level**				
Less than 11 years	612 (76.6)	527 (77.3)	77 (74.0)	8 (61.5)
11 years or more	185 (23.1)	154 (22.6)	26 (25.0)	5 (38.5)
No data	2 (0.3)	1 (0.1)	1 (1.0)	0 (0.00)
**Socioeconomic status**				
Low	715 (89.5)	613 (89.9)	90 (86.5)	12 (92.3)
High	35 (4.4)	29 (4.2)	6 (5.8)	0 (0.0)
No data	49 (6.1)	40 (5.9)	8 (7.7)	1 (7.7)
**Social security in health**				
Contributory	391 (48.94)	325 (47.65)	58 (55.8)	8 (61.5)
Subsidized	408 (51.06)	357 (52.35)	46 (44.2)	5 (38.5)
**Overcrowding**				
Yes	26 (3.2)	22 (3.2)	4 (3.8)	0 (0.0)
No	773 (96.8)	660 (96.8)	100 (96.2)	13 (100.0)
**Number of pregnancies**				
1	382 (47.8)	319 (46.8)	55 (52.9)	8 (61.5)
2	220 (27.5)	194 (28.4)	24 (23.1)	2 (15.2)
3 or more	197 (24.7)	169 (24.8)	25 (24.0)	3 (24.0)
**Gestational week at delivery**				
37	149 (18.7)	124 (18.2)	21 (20.3)	4 (30.8)
38	183 (22.9)	154 (22.6)	27 (26.0)	2 (15.4)
39	227 (28.4)	193 (28.3)	30 (28.8)	4 (30.8)
40	203 (25.4)	181 (26.5)	20 (19.2)	2 (15.4)
41	37 (4.6)	30 (4.4)	6 (5.7)	1 (7.6)
**Maternal morbidity**				
Yes	72 (9.0)	65 (9.5)	6 (5.8)	1 (7.7)
No	727 (91.0)	617 (90.5)	98 (94.2)	12 (92.3)
Total	799 (100.0)	682(85.4)	104 (12.3)	13 (1.6)
**Newborns**	**Total** **n (%)**	**Seropositive** **n (%)**	**Seronegative** **n (%)**	**Equivocal** **n (%)**
**Sex**				
Male	392 (52.2)	341 (51.9)	47 (52.8)	4 (80.0)
Female	359 (47.8)	316 (48.1)	42 (47.2)	1 (20.0)
**Birth weight (grams)**				
3000 or more	483(64.3)	425(64.7)	55(61.8)	3(60.0)
Less than 3000	268(35.7)	232(35.3)	34(38.2)	2(40.0)
Total	751 (100.00)	657 (87.5)	89 (11.8)	5 (0.7)

**Table 2 vaccines-10-00052-t002:** Seropositivity rate for varicella by variables of the pregnant woman and the newborn (umbilical cord). Aburrá Valley, Antioquia, Colombia.

Variable		Pregnant Woman			Newborn		
		Incidence Rate Ratio (IRR)	95% CI	Percentage Change in Seropositivity	Incidence Rate Ratio (IRR)	95% CI	Percentage Change in Seropositivity
Area of residence	Urban	Reference	-		Reference	-	
	Rural *	0.4	0.2–0.7	−60 (−77 to −30)	0.5	0.3–0.9	−50 (−70 to −10)
Socioeconomic status	Low	Reference	-		Reference	-	
	High *	0.4	0.2–0.7	−60 (−77 to −30)	0.5	0.3–0.9	−50 (−70 to −10)
Gestational week at delivery	37	Reference	-		Reference	-	
	38	1.2	0.7–1.8	20 (−30 to 80)	1.1	0.8–1.7	10 (−20 to 70)
	39	1.3	0.9–2.1	30 (−10 to 110)	1.1	0.8–1.8	10 (−20 to 80)
	40	1.4	0.9–2.2	40 (−10 to 120)	1.3	0.8–2.0	30 (−20 to 100)
	41	0.5 *	0.3–1.0	−50 (−70 to 0)	0.6	0.3–1.1	−40 (−70 to 10)
Mother’s age (in years)	<18	Reference	-		Reference	-	
	18 to 34	1.5	0.9–2.5	50 (−10 to 150)	1.4	0.9–2.1	40 (−10 to 110)
	>35	0.6	0.3–1.1	−40 (−10 to 10)	0.7	0.4–1.3	−30 (−60 to 30)
Educational level	<11 years	Reference	-		Reference	-	
	≥11 years *	0.6	0.4–0.8	−40 (−60 to −20)	0.6	0.5–0.8	−40 (−50 to −20)
Birth weight (grams)	≥3000	-	-		Reference	-	
	<3000	-	-	-	0. 8	0.6–1.0	−20 (−40 to 0)

* refers to the value of statistical significance *p* < 0.05.

## Data Availability

The data presented in this study are available on request from the corresponding author. The data are not publicly available due to ethical and privacy issues.

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
