# Peer review of "Seroprevalence of Varicella in Pregnant Women and Newborns in a Region of Colombia"

_vaccines, 2021, doi:10.3390/vaccines10010052_

Round 1

Reviewer 1 Report

Lenis–Ballesteros et al. present a study designed to estimate the seroprevalence of IgG antibodies to varicella zoster virus (VZV)via serological analysis of a cohort of pregnant women and newborns from the Aburrá Valley, Antioquia – Colombia.  The cohort was from women who delivered at eight randomly chosen hospitals.  The study had a straightforward design.  Anti-VZV IgG antibodies were detected in blood samples via ELISA.  Models were designed to identify variables that modify seropositivity.

Weaknesses:   The manuscript is that it is not very innovative.  It is simply a serological survey. However, it has relevance for healthcare in Columbia and perhaps other South American countries.  

Strengths:

1. In pregnant women, seropositivity was 85.8% with seronegativity at 12.6%. The seropositivity of pregnant women was lower for those in rural areas, belonged to the high socioeconomic status and had more education. 

2. In newborns, seropositivity was lower in those who weighed less than 3000 grams. The high seropositivity and seronegativity pattern suggest that preconception consultation and vaccination reinforcement for women of childbearing age according to sociodemographic conditions may prevent VZV prevent infection and complications. Universal vaccination for VZV was established in Columbia in June 2015 as a single dose for children aged one year old covering those born after July 1, 2014. In June 2019, the application of a booster at five years age established as a national guideline.  In this case there would still be a number of individual (pregnant women and newborns) not protected from VZV infection. 

3. Importantly, the authors provide a scientific basis for the need for additional varicella prevention measure that reduce the risk of complications derived from primary infection during pregnancy in pregnant women and thus mitigate the risk of neonatal or congenital varicella syndrome.

The improvement of the manuscript:

The only recommended addition would be to provide a table summarizing other similar VZV serological studies / surveys in Columbia and other South American countries with similar vaccination policies, and comparing those data to the current study.

Author Response

Reviewer 1:
Comment
English language and style are fine/minor spell check required
Answer:
English language and style were revised (see translator's letter)
Comment
The only recommended addition would be to provide a table summarizing other similar VZV serological studies / surveys in Colombia and other South American countries with similar vaccination policies, and to compare those data with the current study.
Answer:
In the introductory section, from lines 52 to 58 on page 2, seroprevalence studies in Latin American populations, from Brazil, Argentina, Chile, the Dominican Republic, Mexico and Colombia are referenced; notably, prior research has not has not explored pregnant women.
In this same section, the following is stated:
"There are no known published studies on seroprevalence in pregnant women or the transfer of antibodies to the newborn to guide the development of preventive programs for maternal, congenital and neonatal varicella in Colombia and Latin America [8 ]" (lines 79-81), a statement that was confirmed again by conducting a complementary literature search in which no articles from South America related to varicella seroprevalence in pregnant women were found.
In the new search, PubMed, SciELO, Science direct, and Bireme search engines were included, finding articles that relate seroprevalence in pregnant women from other regions of the world, such as Egypt, Catalonia, and Europe in general, which were cited in the discussion.

We appreciate this suggestion, but as described, we cannot build the recommended summary table due to the limited scope of prior research. We hope to fill this knowledge gap with this manuscript. 

Reviewer 2 Report

General Comments

In their article, the authors deal with the special problem of Vaticella-Zoster Virus (VZV) infection during pregnancy and after delivery in pregnant women without precedent VZV primary infection. The problem of VZV infection during and after pregnancy seems to be similar dangerous for mother and child as rubella, and therefore pre-pregnancy varicella vaccination in seronegative women seem to be as important as in rubella but is perhaps less paid attention to.

With this background, the authors investigated in a sero-epidemiological study the seroprevalence of IgG antibodies to VZVin several randomly chosen hospitals in a region of Colombia (799 women in late pregnancy, gestational age week 37 or more; additional umbilical cord serum representing the newborn child n=751).

In the introduction the authors report on the general historical development of VZV seroprevalence in Colombia from before vaccination (prevalence 69% in young people aged 1 to 39 years in 2006) as well as since introduction of a VZV vaccination program in 2014 to 2019.

This report seems to be the first one in Colombia on seroprevalence of pregnant women at pre-delivery stage together with umbilical cord seroprevalence, that means in a highly risk group for VZV infection (pregnant women and their children). The results are compared with investigations in other countries at other time periods (Australia 2001, Spain 2007, and Egypt 2019). Seropositivity was still only 86%, seronegativity 13% despite the vaccination strategy since years ago in Colombia. Seroprevalence was influenced by the socioeconomic status, the living area (urban or rural area), and the educational level; in the newborn children also by the body weight at birth. Otherwise, the newborn seroprevalence titre was mainly concordant with that of their mother.

Special Comments

This is an interesting study and is important fort he pre-pregnancy councelling oft he women to get VZV-vaccinated before getting pregnant. Perhaps it should be accentuated that vaccination should be done before and not after start of pregnancy.

Introduction, page 2, lines 45-49: „childhood vaccination" should induce an increase (not a decrease) of he VZV antibody titre; the sentence may be misunderstood in the here given context.

Materials and Methods, page 3, lines 116-120: The authors give three different antibody titres (>9, 9-11, >11); what does the antibody index mean, what is the cut-off being positive for preceding infection? Can the titre level makes a differentiation possible between preceding natural exposure to infection or preceding vaccination?

Author Response

Reviewer 2:
Comment
This is an interesting study and it is important to advise pre-pregnancy women to get vaccinated against VZV before becoming pregnant. Perhaps it should be emphasized that vaccination should be done before and not after the onset of pregnancy.
Answer:
The following sentence is included in the discussion, on page 8, lines 281-286: “under the consideration that vaccination should be done prior to pregnancy”.
Comment
Introduction, page 2, lines 45-49: "infant vaccination" should induce an increase (not a decrease) of the VZV antibody titer, the sentence may be misunderstood in the context given here.
Answer:
This idea was omitted as it did not add relevant information and we clarified that this study corresponds to "a pre-vaccination era" (line 89).
Comment
Materials and methods, page 3, lines 116-120: the authors give three different antibody titers (> 9, 9-11,> 11); What does the antibody index mean? What is the positive cut-off value for the previous infection? Can the titer level make possible a differentiation between natural exposure prior to infection or prior vaccination?
Answer:
What does the antibody index mean?
The antibody index was calculated by multiplying the ratio of the Optical Density (OD) of the sample to the OD of the cutoff serum by 10 (sampleOD/cutoffOD) x 10) (line 119, page 3).
What is the positive cut-off value for the previous infection? Can the titer level make possible a differentiation between natural exposure prior to infection or prior vaccination?
The cutoff value for previous infection is defined as an antibody index greater than 11, as mentioned in the manuscript (line 131-133, page 3).
In the manuscript, it was recognized that "we do not know the history of disease or of vaccination for varicella in pregnant women" (lines 260-261, page 8).
We added: As has been established in the literature, antibody titers do not make any differentiation between natural or vaccine exposure prior to infection and a correlate of protection against the varicella-zoster virus has not been fully established (Wutzler P, Bonanni P, Burgess M, Gershon A, Safadi MA, Casabona G. Varicella vaccination - the global experience. Expert review of vaccines. 2017;16(8):833-
43. https://doi.org//10.1080/14760584.2017.1343669///Habib MA, Prymula R, Carryn S,
Esposito S, Henry O, Ravault S, et al. Correlation of protection against varicella in a randomized Phase III varicella-containing vaccine efficacy trial in healthy infants. Vaccine.
2021;39(25):3445-54. https://doi.org//10.1016/j.vaccine.2021.02.074 ). We assume that the pregnant women in our study were not exposed to varicella vaccination in childhood, since universal vaccination aimed at children only began in 2015 in Colombia (line 123 – 127,
page 3)
